# Heritable gene expression variability and stochasticity govern clonal heterogeneity in circadian period

**K. L. Nikhil**[1,2], **Sandra Korge**[1,2], **Achim Kramer**[1,2]*

**1** Charité Universitätsmedizin Berlin, Freie Universität Berlin and Humboldt-Universität zu Berlin, Laboratory of Chronobiology, Berlin, Germany, **2** Berlin Institute of Health (BIH), Berlin, Germany

* achim.kramer@charite.de

**Data Availability Statement:** Exome sequencing data used in the manuscript are available at https://zenodo.org/ [DOI: 10.5281/zenodo.3876533]. All other relevant data are within the article and its Supporting Information files.

## Abstract

A ubiquitous feature of the circadian clock across life forms is its organization as a network of cellular oscillators, with individual cellular oscillators within the network often exhibiting considerable heterogeneity in their intrinsic periods. The interaction of coupling and heterogeneity in circadian clock networks is hypothesized to influence clock's entrainability, but our knowledge of mechanisms governing period heterogeneity within circadian clock networks remains largely elusive. In this study, we aimed to explore the principles that underlie intercellular period variation in circadian clock networks (clonal period heterogeneity). To this end, we employed a laboratory selection approach and derived a panel of 25 clonal cell populations exhibiting circadian periods ranging from 22 to 28 h. We report that a single parent clone can produce progeny clones with a wide distribution of circadian periods, and this heterogeneity, in addition to being stochastically driven, has a heritable component. By quantifying the expression of 20 circadian clock and clock-associated genes across our clone panel, we found that inheritance of expression patterns in at least three clock genes might govern clonal period heterogeneity in circadian clock networks. Furthermore, we provide evidence suggesting that heritable epigenetic variation in gene expression regulation might underlie period heterogeneity.

## Introduction

The majority of life forms on earth exhibit approximately 24-h (circadian) behavioral and physiological rhythms generated by endogenous timekeeping mechanisms—circadian clocks. In addition to driving such endogenous rhythms, circadian clocks facilitate synchronization of organisms' rhythms to daily and seasonal changes in the environment to enhance their survivability, thereby functioning as an adaptive mechanism [1]. The fundamental basis of circadian rhythm generation across all life forms are cell-autonomous molecular oscillators comprising evolutionarily conserved autoregulatory transcription-translation feedback loops [2]. In higher organisms, such cell-autonomous clocks often function as a network of coupled oscillators that, in unison, drive circadian rhythms [3]. Welsh and coworkers first reported that neurons

**Funding:** Work in AK's lab is supported by grants from the Deutsche Forschungsgemeinschaft to SK (SFB740/D2) and AK (TRR186/P17). Alexander von Humboldt-Stiftung supported NKL's research. The funders had no role in study design, data collection and analysis, decision to publish, or preparation of the manuscript.

**Competing interests:** The authors have declared that no competing interests exist.

**Abbreviations:** ANOVA, analysis of variance; CNV, copy number variant; cps, counts per second; HDAC, histone deacetylase; HSD, honestly significant difference; LCL, long-period clonal line; PC, principal component; PCA, principal component analysis; SAHA, suberoylanilide hydroxamic acid; SCL, short-period clonal line; SCN, suprachiasmatic nucleus; SNP, single nucleotide polymorphism.

within the suprachiasmatic nucleus (SCN; the master pacemaker in the hypothalamus of mammals) are surprisingly heterogeneous in their intrinsic periods of circadian firing pattern [4]. Subsequent studies revealed that such period heterogeneity is not restricted to the SCN but is also observed in mammalian peripheral clock cells [5,6] as well as in *Drosophila* clock cells [7] and plants [8,9]. The ubiquity of this phenomenon suggests that heterogeneity may be functionally relevant for circadian clocks [10–18], thus likely being a substrate for natural selection. Interestingly, the observed period heterogeneity among circadian clock cells within an organism cannot be entirely attributed to functionally different cell types, as cells of the same subtype (clonal cells) also exhibit such variation [5,6]. Clonal heterogeneity or clonal phenotypic variability is common in biology and can stem from various factors such as stochastic changes in the microenvironment, stochastic partitioning of cellular components during cell division, or stochasticity in gene expression [19–25]. In this study, we aimed to explore the possible mechanisms underlying clonal heterogeneity of circadian period in human circadian oscillator cells.

Based on previous reports exploring heterogeneity in other cellular phenotypes, we hypothesized that clonal period heterogeneity in mammalian cells is due to (1) stochastic variation [24,26–28] and/or (2) heritable variation [29–31]. Since the term "stochastic" is used in the context of both nonheritable (external noise and gene expression noise) and heritable gene expression variation (epigenetic stochasticity), for the rest of this manuscript we define "stochasticity" as any nonheritable variation. To test the two above outlined hypotheses, we employed a laboratory selection approach and derived a panel of 25 clonal cell lines (from a common founding culture) exhibiting a range of periods between 22 and 28 h. We observed that the period heterogeneity among progeny clones stemming from a single parent cell is governed by both stochastic and heritable components. Moreover, the extent to which heritable and stochastic components influence circadian period varies between short- and long-period clones. We then measured expression of 20 clock and clock-associated genes in our panel and observed that variation in gene expression of at least three clock genes (transcription factors) might underlie clonal period heterogeneity. Furthermore, we report that the short- and long-period clones are differentially affected by treatment with epigenetic modifier drug and also have different methylation signatures, thus providing preliminary evidence suggesting that epigenetic variation in gene expression regulation might contribute to the heritable basis of clonal period heterogeneity.

## Results

### Both heritable and stochastic components contribute to clonal period heterogeneity

Is the variation in period among individual circadian oscillator cells due to nonheritable stochastic noise? Or is there a heritable component? To test this, we single-cell cloned a "founding culture" of human U-2 OS cells (an established model of peripheral circadian clocks) harboring a *BMAL1*-luciferase reporter construct [32]. Upon reaching confluence, the period of bioluminescence rhythms from these progeny cultures was determined by live-cell bioluminescence recording. We observed a distribution of circadian periods among clones from the founding culture (23.5–27.5 h; Fig 1A top panel). We further repeated this protocol for several "assay generations" by each time selecting short- and long-period clones as "parents" for the successive assay generation (study outline in S1 Fig).

Interestingly, by repeating this protocol for several assay generations, we observed a directional divergence of the progenies' period distributions on either side of the founding culture's distribution (Fig 1A and 1B, S2A Fig). Over the course of the selection protocol, the circadian periods of short- and long-period clonal lines (SCLs and LCLs) diverged from each other and

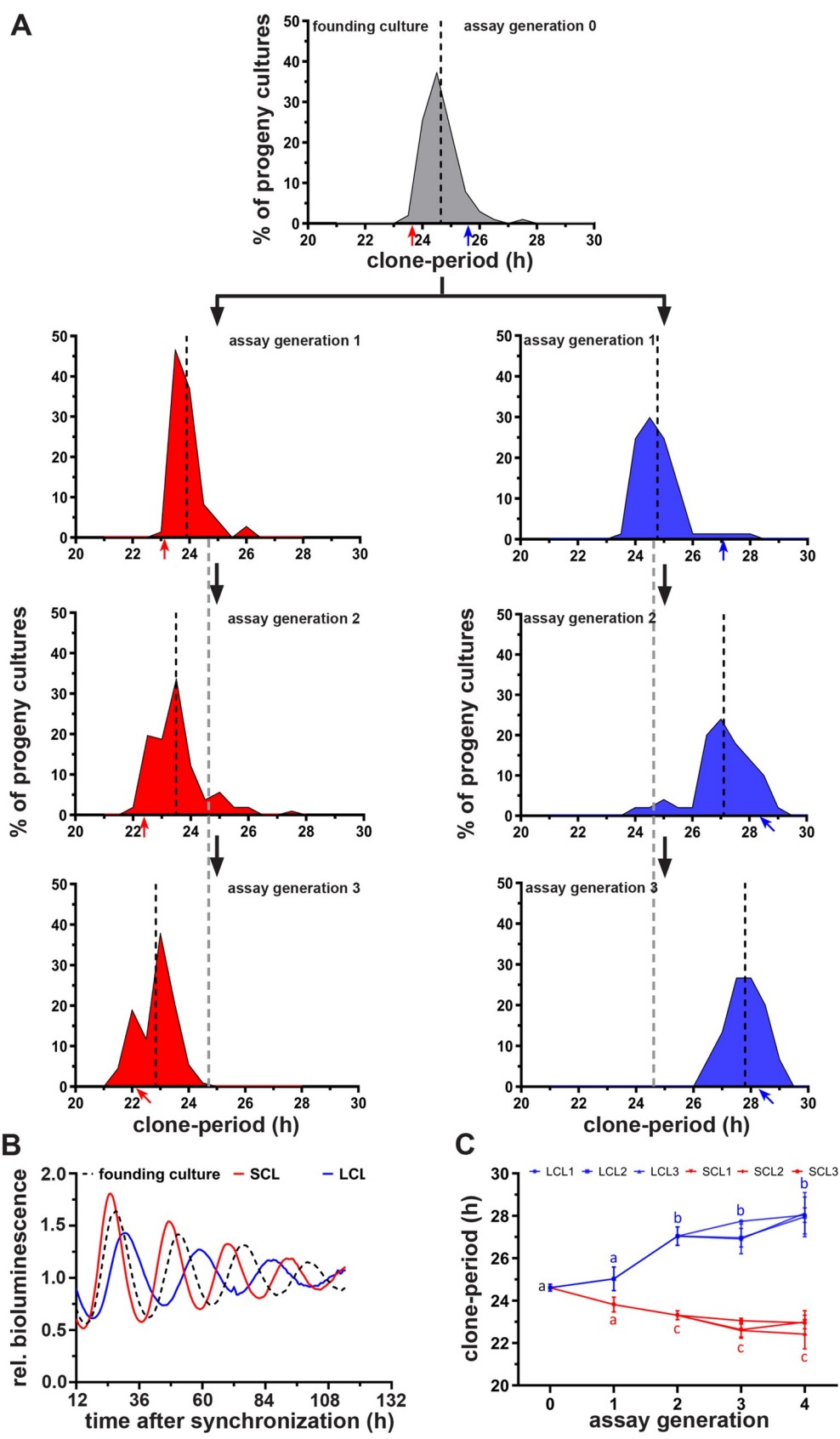

**Fig 1. Both heritable and nonheritable stochastic components contribute to clonal circadian period heterogeneity.**
**(A)** Divergence of circadian period distributions of short-period (red) and long-period (blue) clones from a common founding culture (gray) across multiple assay generations. Dashed black lines depict the mean of respective period distributions. The gray dashed lines extended from assay generation 1 depict mean period of the founding culture (assay generation 0) for visual assessment of the period divergence. Red arrows (short-period clone) and blue arrows (long-period clone) indicate the periods of representative clones selected for the successive assay generation. **(B)** Detrended bioluminescence traces of representative clones from founding culture (dashed line), SCL (red), and LCL (blue). Detrending results in truncation of 12-h data at the beginning and end of time series. **(C)** Mean circadian periods of three representative SCLs (SCL1–3) and LCLs (LCL1–3) across four assay generations. Error bars are SD ($n$ = 3 experiments). Letters a–c depict statistical significance for both interassay and intra-assay generation comparisons. Same letters indicate lack of statistical significance between the average periods of SCLs and LCLs, whereas different letters indicate statistically significant differences in period (ANOVA followed by Tukey's HSD; $p < 0.001$). Underlying data for this figure can be found in S1 Data. ANOVA, analysis of variance; HSD, honestly significant difference; LCL, long-period clonal line; rel., relative; SCL, short-period clonal line.

from the founding culture (Fig 1A–1C). The circadian periods of both SCLs and LCLs diverged significantly by assay generation 2 (Fig 1C; analysis of variance [ANOVA] followed by Tukey's honestly significant difference [HSD], $p < 0.001$). Even though the periods of SCLs and LCLs continued to diverge through assay generations 3 and 4, the difference in periods between assay generations 2 and 4 was not statistically significant for either clones (Fig 1C; ANOVA followed by Tukey's HSD, $p = 0.52$), suggesting that period divergence reached saturation. Thus, the divergence in circadian periods among clonal lines in response to the imposed directional selection suggests that heritable components underlie circadian period heterogeneity.

The mean circadian periods of progeny cultures in every assay generation were similar to their parental cultures, suggesting that the parental circadian period is a very good predictor of the mean progeny period and thus is heritable (S2B Fig). However, progeny cultures continued to exhibit distribution of periods even after three assay generations when divergence of periods had reached saturation (Fig 1A and 1C). Both heritable genetic variation ($V_H$) and nonheritable stochastic/environmental variation ($V_{NH}$) are known to contribute to phenotypic variation ($V_P = V_H + V_{NH}$) observed in populations [33,34]. Heritability estimates help to summarize what proportion of phenotypic variation across individuals derives from variation in heritable components as opposed to variation in nonheritable components and can be calculated as $H^2 = V_H/V_P$, the ratio of heritable genetic variance to the total phenotypic variance, where $H^2$ is termed broad-sense heritability [33,34]. To explore the contributions of heritable and nonheritable components to circadian period heterogeneity, we used variance partitioning by ANOVA to estimate heritable variation ($V_H$) and nonheritable variation ($V_{NH}$) for SCLs and LCLs as described by Lynch and Walsh [34] (see S2C Fig and Methods for more details). In brief, we considered a group of parental SCL and LCL clones and analyzed their respective progeny clones' periods by ANOVA to obtain "between-clone" and "within-clone" variances. Between-clone variance is a measure of variation between progeny clones from different parents and represents variation due to both heritable and nonheritable components. Within-clone variance is a measure of variation within progeny clones of the same parent, and since they are genetically identical, the observed within-clone variation is likely to be primarily due to nonheritable variation (S2C Fig; [34]). Thus, the within-clone and between-clone variances can be used to estimate $V_H$ and $V_{NH}$ (S2C Fig). We analyzed up to 73 progenies from SCLs and LCLs and found that for SCLs at assay generation 0, the $V_H$ and $V_{NH}$ were 0.65 and 0.44, respectively, suggesting that short circadian period has a stronger heritable component compared to nonheritable component and consequently is largely heritable ($H^2 = 0.6$). Interestingly, even at assay generation 0, LCLs had a considerably lower $V_H$ of 0.17 and a relatively higher $V_{NH}$ (0.32) with $H^2$ being 0.34, suggesting that long circadian periods have a weaker heritable component and likely are

more strongly driven by nonheritable stochastic noise. By assay generation 3, $V_H$ of both SCLs and LCLs drastically reduced to 0.08 and 0.09, respectively, and consequently had low heritability (0.15 for SCLs and 0.14 for LCLs), which explains the saturation of divergence in circadian periods over generations (Fig 1C). However, both SCLs and LCLs continued to have higher $V_{NH}$ (0.42 and 0.59, respectively), suggesting that the observed period heterogeneity even after three assay generations is largely due to nonheritable components (Fig 1A).

To briefly assess whether the period differences between the SCLs and LCLs also translate to a different phase of entrainment in the presence of a zeitgeber cycle [35], we measured bioluminescence rhythms of a representative SCL and LCL clones under two different temperature cycles. We observed that indeed the SCL exhibits an advanced phase of entrainment compared to LCL by about 3.2 ± 1.3 h (mean ± SD) under T24 (12 hours of 37˚C and 33˚C each; S3 Fig top panel). This phase-difference further increases to about 5.8 ± 1.7 h (mean ± SD) under T26 (13 hours of 37˚C and 33˚C each; S3 Fig bottom panel) cycles, consistent with theoretical predictions [36].

Taken together, these results indicate that (1) both heritable and stochastic components contribute to clonal period heterogeneity and (2) short circadian periods have a stronger heritable component compared to long periods, which are more strongly noise driven. In addition, results from entrainment to temperature cycles reveal that circadian entrainment properties are conserved even in single-cell clones, thereby underscoring laboratory selection approach as a useful strategy to generate clonal cell populations with divergent circadian clock traits that can aid chronobiology studies, which are sometimes limited by the potential pleiotropic effects encountered using clock mutants.

## Precision of circadian rhythm decreases with increasing period

Several earlier studies reported correlation of circadian period with rhythm precision (a measure of intercycle period stability). They suggested that precision of circadian rhythms is high when the circadian period is closer to 24 h and decreases as period deviates from 24 h, but mostly at the organismal level and within the pacemaker SCN neurons [37–42]. In addition, we find that long circadian periods are largely noise driven as compared to short periods. Thus, we tested (1) whether correlation of period with rhythm stability is also observed in peripheral oscillator cells and (2) whether the largely noise driven long-period cells exhibit stable circadian rhythms. To this end, we measured the SD of intercycle (peak-to-peak) period in our clonal lines.

We observed a significant positive correlation of intercycle period variation with the clone period (Spearman $r = 0.51$, $p < 0.0001$; Fig 2A). Clones with shorter circadian periods had a higher rhythm stability (lower SD of intercycle period), which reduces as period lengthens (Fig 2A). Interestingly, not all long-period clones exhibit reduced rhythm stability, and the interclonal variation in rhythm stability appeared to be higher among long-period clones compared to their short-period counterparts (Fig 2A).

These results suggest that rhythm stability is indeed associated with clock period even in peripheral cellular oscillators, with long-period oscillators having a higher propensity to exhibit reduced rhythm stability. However, the nature of association between rhythm stability and clock period does not agree with previous reports, as will be discussed later.

## Clonal period heterogeneity is ubiquitous and likely not caused by genetic polymorphisms

The U-2 OS cells used in our study are osteosarcoma-derived cells, and cancer-derived cell lines are known to have a higher mutation propensity [43,44]. Therefore, we asked whether

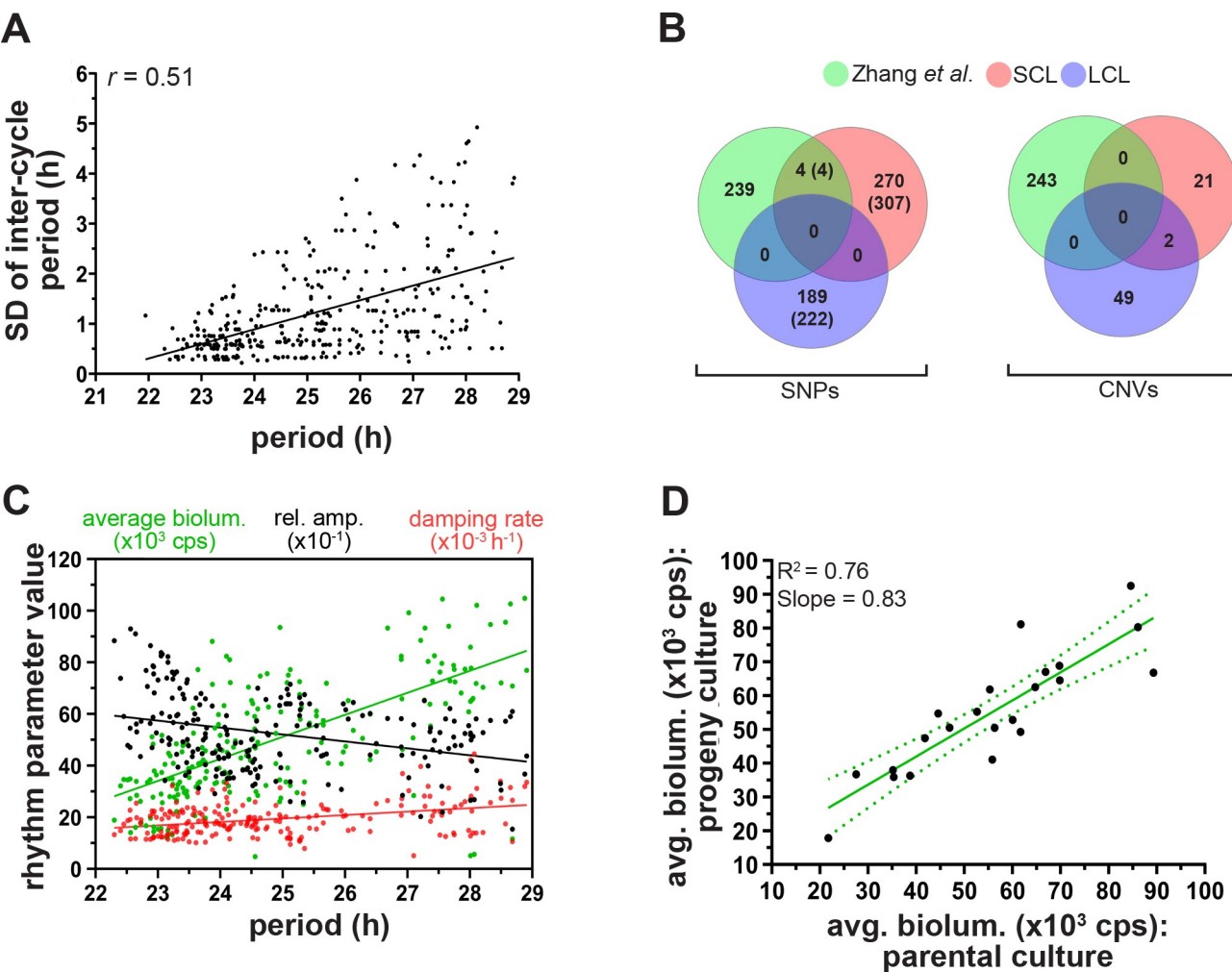

**Fig 2. Rhythm stability is associated with period, and clonal period heterogeneity does not stem from polymorphisms. (A)** Correlation of SD of intercycle (peak-to-peak) period with clonal circadian period indicating a reduction in rhythm stability with increasing clock period. **(B)** Venn diagrams depicting overlap of SNPs and CNVs identified among SCLs and LCLs with the 243 period modifier genes reported by Zhang and colleagues [46]. For SNPs, numbers within brackets indicate the total number of SNPs identified, and numbers otherwise represent the total number of genes harboring the identified SNPs. **(C)** Correlation of rhythm parameters—average bioluminescence ("biolum."; green), relative amplitude ("rel. amp."; blue), and damping rate (red)—with clonal circadian period. **(D)** Regression of average bioluminescence ("avg. biolum.") of parental clones over progeny clones, indicating that parental average bioluminescence is a very good predictor of progeny bioluminescence. Green solid line is the linear regression fit with its 95% CI (green dotted line). Underlying data for (A), (C), and (D) can be found in S1 Data. Data used for (B) are available at https://zenodo.org/ (DOI: 10.5281/zenodo.3876533). CNV, copy number variant; cps, counts per second; LCL, long-period clonal line; SCL, short-period clonal line; SNP, single nucleotide polymorphism.

the observed circadian period heterogeneity is specific to U-2 OS cells (and possibly other hypermutable cells) due to genetic instability or whether it is ubiquitous and can be extrapolated to cell types in other species. To test this, we imposed a similar artificial selection protocol for one assay generation on mouse cells NIH 3T3 (an immortalized murine fibroblast cell line with a much lower spontaneous mutation propensity [45]) expressing a *BMAL1* promoter–driven luciferase. Similar to U-2 OS cells, single-cell cloning of NIH 3T3 cells from founding culture resulted in a distribution of periods 22.80 ± 0.7 h (mean ± SD) at assay generation 0 (S4 Fig). We then selected short- and long-period clones from this distribution and single-cell cloned these for the next assay generation. As in U-2 OS cells, we observed that by assay generation 1, the average period of short clones reduced by 1.2 h to 21.74 ± 0.2 h (mean ± SD),

whereas that for long-period clones increased by 0.5 h to 23.4 ± 1.1 h (mean ± SD; S4 Fig). These results suggest that clonal period heterogeneity is likely not due to hypermutation in cancer cells but appears to be a ubiquitous phenomenon.

To nevertheless test whether period divergence over assay generations might be caused by differential accumulation of period-changing mutations in SCLs versus LCLs, we sequenced the exomes of three representative SCLs and LCLs each along with the founding culture. When referenced against the human genome, we found an average of 168,982 ± 7,493 (SD) single nucleotide polymorphisms (SNPs) across the seven sequenced clones, thus underscoring the hypermutable genome of U-2 OS cells. However, when we compared SNPs in SCLs and LCLs with those in the founding culture, we observed that about 99.8% of all SNPs in SCLs and LCLs were already present in the founding culture. We then compared SNPs shared between all three SCLs and LCLs and identified 311 SNPs across 274 genes in SCLs and 222 SNPs across 189 genes in LCLs (S1 Table). These SCL- and LCL-specific SNPs are polymorphisms that likely arose in our clones during the course of the selection and may, in principle, underlie the observed period differences. Zhang and colleagues [46] executed a genome-wide RNAi screen using U-2 OS cells and reported 243 genes, which upon knockdown resulted in circadian period changes (S1 Table). To test whether the SCL- and LCL-specific SNPs that we identified may have a causal link with circadian period, we compared the list of period modifier genes reported by Zhang and colleagues [46] with the genes harboring the identified SNPs in our clones. We found that SCLs had four common SNPs in four period modifier genes (*ZNF91*, *SART3*, *GFAP*, *ZMAT3*) and one common SNP in *CLOCK*, whereas none of the common genes harboring LCL-specific SNPs were found to be period modifiers (Fig 2B). Annotation of the five identified SNPs in SCLs revealed that none of the SNPs caused an amino acid change in the corresponding proteins. One SNP (in *ZNF91*) was a synonymous variant, three SNPs (in *SART3*, *CLOCK*, and *ZMAT3)* were intronic variants, and one SNP (in *GFAP*) was a 3′-UTR variant (S2 Table). In addition, none of these SNPs were present in annotated splice junctions, promoters, enhancer elements, or any experimentally verified transcription factor, RNA binding protein, or microRNA binding sites, thus suggesting that these SNPs are unlikely to be causally linked to short period in SCLs (S2 Table). To further test this, we compared the expression levels of these between three representative SCLs and LCLs and did not find any significant differences in their expression levels (S5 Fig; expression of *CLOCK* will be discussed below). These data also indicate that genetic polymorphisms acquired during the selection process are unlikely to contribute to the short or long circadian periods in our clones. In addition, we analyzed copy number variants (CNVs) and did not find any CNV commonly shared by all three SCLs or LCLs. When CNVs across all three clones were pooled together, we found 23 CNVs across the three SCLs and 51 CNVs across the three LCLs with at least one copy gain or loss (S3 Table). However, none of these genes were reported by Zhang and colleagues [46] as period modifier genes either (Fig 2B; S3 Table). Interestingly, among the six clones analyzed, four clones (two SCLs and two LCLs) had a copy number loss in *CLOCK* (S3 Table), but since they were commonly shared between the two period clones, we conclude copy number loss in *CLOCK* is unlikely to be causally linked to period differences between SCLs and LCLs.

Altogether, these results further indicate that clonal period heterogeneity is a ubiquitous phenomenon and is unlikely to be due to genetic polymorphisms.

## Differential expression of E-Box-associated factors may underlie clonal period heterogeneity

Having established that the heritable components underlying period differences observed between SCLs and LCLs are likely not due to accumulated polymorphisms, we further aimed

to explore the heritable basis of period heterogeneity. During the course of our experiments, we observed that SCLs and LCLs consistently exhibited differences in average bioluminescence (average bioluminescence counts across a circadian cycle). LCLs had higher average bioluminescence compared to SCLs (S2A Fig). This encouraged us to test correlation of circadian period with other parameters, and we observed a strong significant positive correlation of average bioluminescence with period (Spearman $r = 0.65$, $p < 0.0001$), but not to relative amplitude (Spearman $r = −0.32$, $p < 0.0001$) and damping rate (Spearman $r = 0.28$, $p < 0.0001$), which only weakly correlated with period (Fig 2C). Furthermore, average bioluminescence of parental clones was found to the best predictor of the respective progeny values ($R^2 = 0.76$; Fig 2D), whereas relative amplitude ($R^2 = 0.04$; S6A Fig) and damping rate ($R^2 = 0.40$; S6B Fig) were only poor predictors.

Based on the above-described results, we reasoned that average bioluminescence can, in principle, serve as a proxy for the average expression of the underlying gene (*BMAL1*) and thus hypothesized that variation in average gene expression of circadian clock genes might underlie clonal period heterogeneity. To test this prediction, we used the NanoString multiplex platform to measure the average expression levels of 20 clock and clock-associated genes (S4 Table) across our panel of 25 clones exhibiting periods spanning 22–28 h. To do so, we carefully plated cells to avoid any inadvertent synchronization due to handling, let the cells grow for 6 days without any medium change, and isolated RNA at 144.5 h post plating. We observed that when handled this way, the cell population was essentially arrhythmic by day 6, with relative amplitudes reducing by approximately 95% compared with day 1 amplitude (see Methods; S7A–S7C Fig; S1 Text).

When then analyzing the expression levels of the 20 clock and clock-associated genes (S4 Table) at 144.5 h post plating across our panel of 25 clones, not surprisingly, we observed a high cross-correlation in expression of the measured genes (S8A Fig). This is likely due to the high interconnectivity in the circadian clock molecular loop, wherein expression changes in one gene may drive expression changes in multiple other genes. We therefore subjected the dataset to principal component analysis (PCA), which is a useful technique for analyzing such high-dimensional correlated data. PCA facilitates reduction of data dimensionality by transforming a number of correlated variables (genes in this case) into uncorrelated principal components (PCs), which are linear combinations of the variables (genes) each having a different contribution to the PC. By identifying relevant PCs, one can further look at what variables (genes) have the most influence on the PC, thus helping identify genes contributing the most to the observed heterogeneity in periods. Based on the Broken Stick model (see Methods and [47]), we retained the first two PCs, which collectively explained 70.2% of the variance in period (Fig 3A). Interestingly, the first two PCs also clustered the panel of clones into three categories of short- (22.3–23 h), intermediate- (23.8–26.9 h), and long-period (27.6–28.2 h) clones (Fig 3B). PC1 clustered the clones into two groups: (1) intermediate periods and (2) the rest including both short- and long-period clones (nonintermediate). In contrast, PC2 appeared to be important for the three observed clusters (Fig 3B). Since any given PC is a linear combination of all genes with different coefficients (a measure of magnitude of a gene's influence on a PC), we use the $\cos^2$ values (Fig 3C) and contributions of genes to PC2 (S8B–S8D Fig), both of which are measures to assess which genes have the most influence on a PC. We shortlisted the top 25% of the candidate genes (*ARNTL2*, *BHLHE40*, *DBP*, *NR1D2*, *PER2*) that we hypothesized to largely account for the period heterogeneity in our panel of clones.

Hierarchical clustering based on expression of the five shortlisted candidate genes clustered clones based on periods similar (with one exception) to that by the first two PCs (Fig 3B). The amalgamation schedule suggested a possibility of three clusters (red, blue, and black dashed rectangles, Fig 3D), which was also in agreement with the optimal cluster number reported by

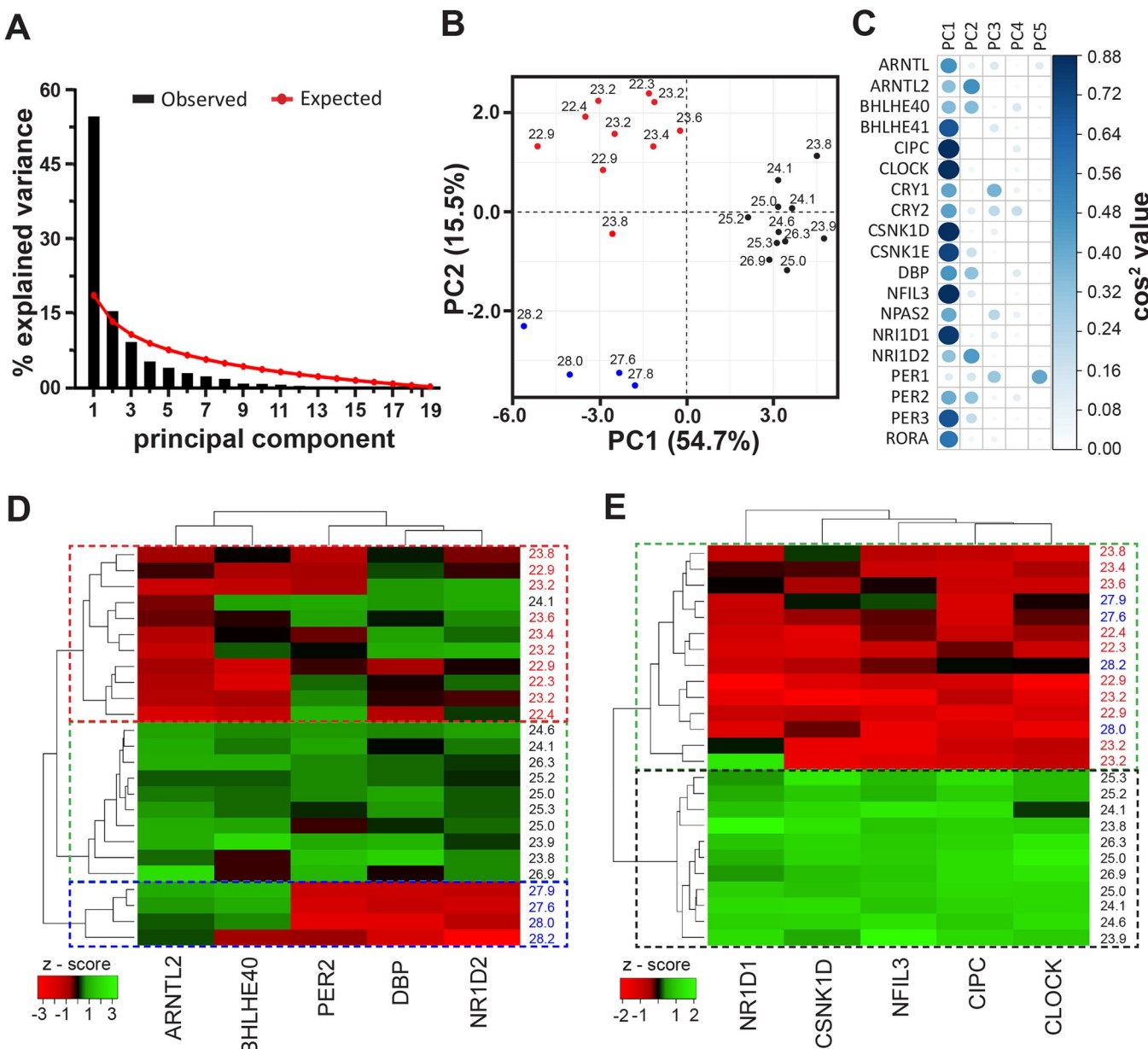

**Fig 3. Inheritance of clock-gene expression patterns might govern clonal period heterogeneity. (A)** Scree plot depicting the percentage of variance explained by the 19 PCs (black bars) and the expected values based on the Broken Stick model (red line). **(B)** Factor map of individual clones plotted across PCs 1 and 2 reveals that the first two PCs cluster the clones in three clusters of short- (red), intermediate- (black), and long-period (blue) clones. **(C)** Cos$^2$ values (a measure of the extent of influence of a gene on the PC) of the 19 genes for PCs 1–5. The color and size of circles represent the magnitude of cos$^2$ value. **(D)** Hierarchical clustering based on the expression of five genes selected from PC2. With the exception of one clone, all others clustered into three groups of short, intermediate, and long clones (red, black, and blue dashed rectangles, respectively). **(E)** Hierarchical clustering based on the expression of five genes from PC1 resulted in two clusters: (1) intermediate period (black dashed rectangle) and (2) short and long period (green dashed rectangle). Color coding of clones in (D) and (E) is the same as in (B). Underlying data for this figure can be found in S1 Data. PC, principal component.

five different indexes (S9A–S9E Fig). Clustering-based heat map revealed that the expression of *ARNTL2* and *BHLHE40* correlated positively with the circadian period, and *DBP* and *NR1D2* correlated negatively (Fig 3D, S10A Fig). We also similarly shortlisted top 25% genes from PC1 (*NR1D1*, *CLOCK*, *CSNK1D*, *CIPC*, and *NFIL3*), and as expected, we observed that these genes were not sufficient to discriminate the short and long periods and resulted in just

two clusters—intermediate and nonintermediate (Fig 3E, S9F–S9J Fig). Interestingly, all five genes from PC1 have higher expression in "intermediate-period" clones, and their expression reduces as the period deviates from "intermediate" (Fig 3E, S10B Fig). We compared these genes with the CNVs identified earlier and did not find any of the clones that harbored a CNV for the above assayed genes (except *CLOCK*), thereby ruling out the possibility that the observed gene expression differences is due to polymorphisms. As mentioned earlier, both SCLs and LCLs had copy number loss for *CLOCK*, and this explains the reduced gene expression in both sets of clones (Fig 3E). Furthermore, reduced expression of *CLOCK* in both SCLs and LCLs suggests that the intronic SNP identified in SCLs is unlikely to be causally linked to period differences between the clones. Thus, we hypothesized that changes in expression of PC2 genes are likely to underlie circadian period heterogeneity in our clones.

If differences in expression of the shortlisted PC2 genes govern period heterogeneity, then depletion of these genes should result in large period change, whereas depletion of those from PC1 should not have a significant effect on period. Specifically, based on their expression patterns (Fig 3D), knockdown of *ARNTL2* and *BHLHE40* should shorten the circadian period, whereas *DBP* and *NR1D2* knockdown should result in period lengthening. To test this, we used RNAi-mediated knockdown of the shortlisted genes in three short-, two intermediate-, and three long-period clones (based on clustering in Fig 3D) and studied the effect on circadian period. Indeed, we observed that knockdown of *NR1D2* resulted in significant period lengthening across all clones, whereas *BHLHE40* and *ARNTL2* knockdown resulted in significant period shortening (mixed-model ANOVA followed by Tukey's HSD; $p < 0.00001$; Fig 4A and 4B). *NR1D2* knockdown had the largest effect on period, significantly higher compared to all other genes across both the PCs, followed by *BHLHE40*, which was similar to *ARNTL2* and had a significantly higher effect on period compared to all other genes. Knockdown of none of the other genes across both PCs resulted in a period change significantly differing either from zero (one-sample *t* test, $p > 0.05$) or from each other (mixed-model ANOVA followed by Tukey's HSD; $p > 0.05$; Fig 4A and 4B). Accordingly, we observed that the average absolute period change upon knockdown of PC2 genes was significantly higher than that by PC1 genes (Fig 4C).

Taken together, these results suggest that differential expression of *NR1D2*, *BHLHE40*, and *ARNTL2* likely underlies clonal heterogeneity in circadian period.

## Epigenetic regulation might underlie altered gene expression patterns associated with clonal period heterogeneity

Having observed that clonal period heterogeneity is associated with altered gene expression patterns, we next asked, What drives such altered expression among clonal cells? We dismissed the possibility of genetic polymorphism as being unlikely (see above and Discussion) and hypothesized that epigenetic variation might account for the observed differences in gene expression patterns among clonal lines.

To this end, we treated all 25 clonal cell populations in our panel with the commonly used epigenetic modifier suberoylanilide hydroxamic acid (SAHA) and studied the effect of the treatment on clone period. SAHA is a class I and class II histone deacetylase (HDAC) inhibitor that up-regulates gene expression by multiple mechanisms [48]. We reasoned that if reduction in expression of the identified subset of genes across our clonal panel is due to epigenetic suppression (in this case, acetylation status), treatment with SAHA should up-regulate the expression of these genes, thereby lengthening and shortening the circadian period in short- and/or long-period clones, respectively.

Interestingly, we observed that treatment with SAHA differentially influenced the short-, intermediate-, and long-period clones. SAHA treatment resulted in a significant period

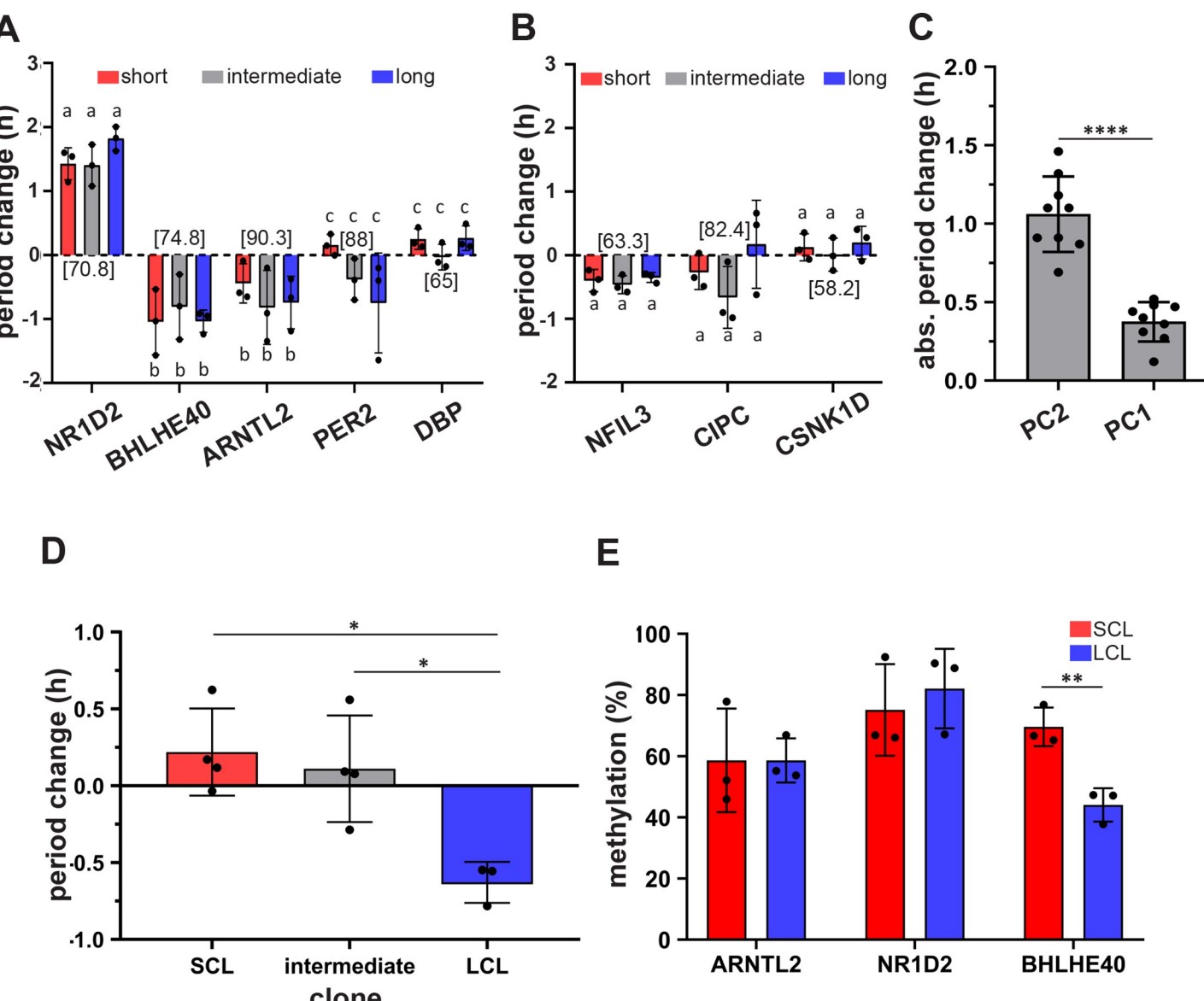

**Fig 4. Epigenetically regulated expression of E-Box-associated factors may govern clonal period heterogeneity.** Period change (compared to nonsilencing scrambled shRNA control) upon knockdown of the **(A)** five PC2 genes and **(B)** three PC1 genes for the short- (red), intermediate- (gray), and long-period (blue) clones. Bars with different letters indicate significant differences ($p < 0.05$), and bars with the same letter are not significantly different from each other (mixed-model ANOVA followed by Tukey's HSD). Numbers within square brackets indicate the knockdown efficiency of the respective gene. **(C)** Averaged absolute ("abs.") period change across all clones upon knockdown of genes from PC2 and PC1. **(D)** Period change (compared to vehicle control) upon treatment of short- (red), intermediate- (gray), and long-period (blue) clones with HDAC inhibitor SAHA (1.6 μM). **(E)** Methylation percentage of CpG islands proximal to the transcription start site of the three genes—ARNTL2, NR1D2, and BHLHe49. For all panels in this figure, $n$ = 3–4 experiments and error bars are SD ($^*p < 0.05$; $^{**}p < 0.001$; $^{****}p < 0.0001$). Underlying data for this figure can be found in S1 Data. ANOVA, analysis of variance; HDAC, histone deacetylase; HSD, honestly significant difference; LCL, long-period clonal line; PC, principal component; SAHA, suberoylanilide hydroxamic acid; SCL, short-period clonal line; shRNA, short hairpin RNA.

shortening in the long-period clones (ANOVA followed by Tukey's HSD, $p < 0.05$; Fig 4D), whereas the magnitude of period change in short- and intermediate-period clones did not differ from each other (ANOVA followed by Tukey's HSD, $p = 0.85$) or from zero (one-sample $t$ test, $p > 0.05$; Fig 4D).

In addition, since the three PC2 genes (*NR1D2*, *BHLHE40*, and *ARNTL2*) whose expression strongly correlated with circadian period in our clones and upon knockdown had significant

impact on period, we asked whether methylation may regulate the observed differential expression of one or more of these genes among our clones. To study this, we used three short- and three long-period clones and assessed the methylation status of CpG islands proximal to transcription start sites of these genes. In accordance with reduced *BHLHE40* expression in the short-period clones (Fig 3D), we observed that the CpG island upstream of *BHLHE40* was significantly more methylated in the short-period clones compared to that in long-period clones (*t* test, $p < 0.01$; Fig 4E), whereas the clones did not differ in their methylation of CpG islands upstream of *ARNTL2* and *NR1D2* genes (Fig 4E).

The possible reasons for the differential effects of SAHA treatment on short- and long-period clones will be discussed later, but taken together, these results provide preliminary evidence suggesting that differential epigenetic regulation of circadian clock–gene expression might underlie clonal period heterogeneity.

## Discussion

We used human U-2 OS cells to investigate whether period heterogeneity in circadian clock network stems from stochastic (nonheritable) components or whether it has a heritable component. We employed a laboratory selection protocol to select for clonal cell lines exhibiting short and long circadian periods (from a common founding culture) through which we derived a panel of 25 clonal cell populations exhibiting circadian periods between 22 and 28 h.

We observed that, in response to imposed selection, circadian periods of SCLs and LCLs continued to diverge from that of the founding culture and saturated over four assay generations (Fig 1A–1C; S2B Fig), suggesting that period heterogeneity in clonal populations is not entirely stochastically driven, but has a considerable heritable component. Interestingly, despite the saturation in period divergence, SCLs and LCLs continued to exhibit heterogeneity in circadian periods (Fig 1A), suggesting that stochastic (nonheritable) variation also contributes to period heterogeneity. We estimated the contribution of heritable and nonheritable components by variance partitioning. We found that although stochastic variation contributes to both short and long periods, short circadian period is more heritable compared to long period, which is largely stochastically driven, in agreement with a recent report on immortalized mouse ear fibroblasts [49]. These findings strongly suggest both heritable and stochastic components govern circadian period heterogeneity.

U-2 OS cells used in our study are a cancer-derived osteosarcoma cell line known to have high mutation propensities [43,44]. Therefore, we asked whether the heritability of observed period differences between short- and long-period cells are due to genetic polymorphisms and therefore specific to U-2 OS (or other cancer) cells or whether period heterogeneity is ubiquitous and can be extrapolated to other species as well. To address these questions, we sequenced exomes of short- and long-period clones along with the founding culture, and our analysis of SNPs and CNVs revealed that these random polymorphisms are unlikely causing the observed period differences between the clones (Fig 2B). In addition, we imposed a similar selection protocol using mouse NIH 3T3 cells, an immortalized murine fibroblast cell line with a much lower number of spontaneous mutations [45], and also found a substantial period divergence of the clones after selection (S4 Fig). Taken together, our results suggest that period heterogeneity is (1) driven by both heritable and nonheritable components, (2) unlikely to be due to genetic polymorphisms, and (3) a ubiquitous phenomenon. This raises an interesting question: why would natural selection favor the evolution of heritable mechanisms to drive period heterogeneity over entirely stochastically driven heterogeneity? We hypothesize that, although period heterogeneity can be functionally beneficial [10–18], very large heterogeneity can negatively influence clock functionality as well [12,13,16]. Entirely stochastically driven

heterogeneity can potentially lead to very large variation in intercellular/oscillator period, which can be detrimental, whereas heritable mechanisms may impose phenotypic constraints [50], within which period heterogeneity can be maintained and thus favored by natural selection. Recently, it has been proposed that isolating fibroblasts from humans and assaying their circadian period might provide a reasonable estimate of the individual's behavioral period [51]. As we report in this study, progeny clones of a single parental clone can differ in their periods, the average period of the progeny clones always resembles the parent clones' period (Fig 2D). Therefore, assaying multiple cells from an individual and estimating their average period is very important for measuring of the individual's period.

Reports by Aschoff [37] as well as Pittendrigh and Daan [38] led the latter to hypothesize that circadian rhythms are most stable as the endogenous period (τ) approaches 24 h, whereas deviation of τ from 24 h leads to reduction in rhythm stability and provided an evolutionary reasoning for such a hypothesis. This observation of rhythm stability being the lowest at a particular τ has since been substantiated by studies on various species [39,41,42], including the pacemaker SCN neurons [40], but others have reported either weak or absence of such trends [52,53]. To assess whether such correlation is observed mostly at the level of organisms and central pacemaker neurons or is a general property even for peripheral oscillator cells, we measured the intercycle variation in period across our clonal populations. We observed that the rhythm stability decreases (SD of intercycle period increases) with increasing clone period (Fig 2A). Moreover, clones with longer periods exhibited a higher interclonal variation in rhythm stability compared to short-period clones, thereby suggesting that longer-period clones have a higher propensity of exhibiting unstable rhythms (Fig 2A). Long-period mutant in *Neurospora crassa* exhibit higher amplitude and are robust compared to short-period mutants who have lower amplitudes [54]. Similarly, long-period mutants of Syrian hamsters also exhibit higher amplitude whereas short-period mutants have lower amplitude and exhibit reduced rhythm stability [52]. Therefore, it is conceivable that observed reduction of rhythm stability in our long-period clones may not be causally associated with circadian period but may be a consequence of reduced rhythm amplitude (Fig 2C) and increased stochastic noise. Our results highlight that rhythm stability is correlated with circadian period, but it is not in agreement with previous studies that have reported increased rhythm stability at a particular τ. It is possible that the previously reported correlations are restricted to the central clock (at least in higher organisms with central and peripheral clock architecture), since the stability of behavioral rhythm is more likely to be the direct substrate for natural selection [38].

Over the course of our experiments, we observed that long-period clones often exhibited higher average bioluminescence compared to the short-period clones (Fig 2C; S2A Fig) and further analysis revealed that parent average bioluminescence was a good predictor of progeny bioluminescence, but this was not the case for either relative amplitude or the damping rate (Fig 2D, S6A and S6B Fig). We reasoned that average bioluminescence could serve as a proxy measure for *BMAL1* expression and hypothesized that variation in gene expression of one or more clock and clock-associated genes might underlie period heterogeneity. To further explore this, we measured average expression of 20 circadian clock and clock-associated genes (S4 Table) across all 25 clones in our panel. By employing PCA, we identified five candidate genes (*ARNTL2*, *BHLHE40*, *DBP*, *NR1D2*, and *PER2*) that grouped clones into three distinct clusters—short, intermediate, and long periods (Fig 3A–3D). Furthermore, we observed that knockdown of three of the shortlisted candidates—*NR1D2*, *BHLHE40*, and *ARNTL2*—had the significantly changed periods in short- and long-period clones whereas knockdown of other genes, including those from PC1 (Fig 3E), had little or no effect on period (Fig 4A–4C). It is noticeable that individual knockdown of the genes resulted in small-magnitude period changes that cannot entirely account for period differences between the short- and long-period clones

(Fig 4A), suggesting that that period heterogeneity is likely a multigene trait involving a consortium of multiple medium-effect genes. Notably, all three above-mentioned genes are transcription factors that are either regulated by and/or act on E-boxes [55–61], suggesting that modulation of clock period is complex and governed by interaction between the interconnected molecular clock loops coupled by E-box-associated transcription factors [62,63].

Intriguingly, in contrast to the above-discussed genes, we find another category among the assayed genes that exhibit an inverted-U-shaped relationship with period (Fig 3E, S10B Fig). The expression of these genes is high in clones with intermediate periods (23.8–26.9 h) and is drastically reduced in clones with periods deviating from the intermediate range. Furthermore, our knockdown studies also confirm that expression patterns of these genes are not causal to but likely a response/consequence to period variation (Fig 4B and 4C). Such inverted-U-shaped responses (hormesis) is observed in various biological systems and is regarded as a regulatory/homeostatic mechanism to prevent very large deviations of cellular/organismal phenotypes from their optimal range [64–66]. As discussed earlier, since a higher degree of period heterogeneity can be detrimental to the circadian clock network, we hypothesize that although there are mechanisms within the clock circuitry that promote period heterogeneity, the network might also harbor hormesis-based mechanisms, which impose constraints on the range of period that the circadian clock can exhibit [64–66]. Such mechanisms may also explain why we observe a saturation of period divergence after assay generation 2 (Fig 1C).

Although evidence thus far strongly suggested that period heterogeneity is driven by differences in clock-gene expressions, and having ruled out the possibility of genetic polymorphisms, we asked, What is the source of these expression differences? Epigenetic variation–driven gene expression differences often underlie phenotypic heterogeneity in many life forms, including clonal populations [15,22,67–70]. Therefore, we hypothesized that differential epigenetic regulation of gene expression may underlie period heterogeneity. To test this, we studied the effect of a HDAC inhibitor SAHA treatment on the circadian period across our clones. Interestingly, we find that treatment with SAHA significantly shortens the period in long-period clones with little or no effect on the short and intermediate clones (Fig 4D). Although the magnitude of period change upon SAHA treatment is relatively small, the differential effects of SAHA on short- and long-period clones are intriguing and suggest that mechanisms underlying short and long circadian periods might be entirely different and not mechanistic opposites of each other. Furthermore, the small magnitude effect of SAHA may be also due to (1) genome-wide effects of SAHA up-regulating other genes also that in turn negatively influence the change in period, thus resulting in overall small magnitude effect on period change, and/or (2) SAHA being a broad-spectrum HDAC inhibitor that promotes up-regulation of genes by acetylation, whereas other epigenetic mechanisms that might also regulate gene expression in our clones are not targeted by this treatment. Therefore, we also assessed methylation of CpG islands proximal to transcription start sites of the three genes *ARNTL2*, *NR1D2*, and *BHLHE40*, which had maximal effect on period upon knockdown (Fig 4A). We observed that CpG islands upstream of BHLHE40 were significantly more methylated in short-period clones compared to long-period clones, which correlates with the reduced BHLHE40 expression in short-period clones (Fig 3D). Since BHLHE40 acts as inhibitor of E-box-mediated transcription [71], it is possible that the expression levels of the other PC2 genes (*ARNTL2*, *PER2*, *NR1D2*, *DBP*) are directly regulated by BHLHE40. Together, these results suggest that different epigenetic mechanisms regulate gene expression differences and contribute to heritable components of period heterogeneity.

Results from studies over the past decade suggest that cell cycle and circadian clock may be coupled, with the consensus that cells that divide faster have shorter circadian periods, whereas the long-circadian-period cells divide slower [5,72–74]. If such coupling also exists in U-2 OS

cells, then it is likely that selection for short- and long-period clones can also lead to correlated selection for higher and lower proliferation rates, respectively. To further explore this, we assayed proliferation rates of six clones (three representative short- and long-period clones) from our study by plating them at two different densities (1,000 cells/well and 5,000 cells/well) and measuring proliferation rate across 7 consecutive days using colorimetric assay. We did not find any difference in proliferation between short- and long-period clones when plated at 1,000 cells/well; however, when plated at 5,000 cells/well, we observe a trend toward long-period clones proliferating faster than short-period clones—directly opposite to previous reports (S11 Fig). Although we do not have an explanation for these counterintuitive results, we speculate that (1) the proliferation differences between short- and long-period clones is not associated with period differences and that circadian clock–cell cycle coupling may be absent in cancer cell lines such as U-2 OS cells as also reported for Lewis lung carcinoma cells [75] and immortalized rat fibroblasts [76]. (2) Alternatively, the negative correlations between circadian period length and cell division rate observed in fibroblasts may be density dependent in U-2 OS or other cancer cells and may require further studies using different cell lines to validate this. The latter would provide an interesting avenue for future studies to explore whether period heterogeneity stems from mechanisms underlying heterogeneity in proliferation rates, which is also governed by both stochastic noise and epigenetic variation [77], thus sharing common heritable basis.

In conclusion, we report that both heritable and stochastic components govern clonal heterogeneity of circadian periods and epigenetically regulated differential expression of E-box-associated transcription factors might govern period heterogeneity. Furthermore, conserved entrainment properties even in our single-cell clones underscore laboratory selection as a useful strategy to generate clonal panel exhibiting a range of circadian phenotypes that can aid interesting chronobiology studies.

## Methods

### Cell culture and selection protocol

Cells used in this study included U-2 OS (human, ATCC # HTB-96) and NIH 3T3 (mouse, ATCC # CRL-1658) cells, which stably expressed firefly luciferase from a 0.9-kb *BMAL1* promoter [32]. All cells were cultured and maintained in DMEM containing 10% fetal bovine serum and antibiotics (100 U/ml penicillin and 100 μg/ml streptomycin).

For selection of clones with different circadian periods, U-2 OS cells from "founding culture" expressing a circadian period of 24.6 ± 0.16 h (mean ± SD) were plated as single-cell clones in 96-well "parent plates" and grown to confluency. On reaching confluency, a replicate "assay plate" was established from every parent plate by splitting cells. Bioluminescence rhythms of clones in assay plates were recorded (see below for recording protocol) and those exhibiting short or long periods (tails of the period distribution) were selected. We observed that in some clones, the period of bioluminescence rhythms was not reproducible when measured again; therefore, every clone was recorded two to three times and only those that consistently exhibited shot/long periods were selected. Following the selection of clones, corresponding clones from the parent plate were again single-cell cloned in 96-well plates by serial dilution (later confirmed visually through microscope by checking every plated well), and the procedure was repeated for four assay generations by selecting short- and long-period clones every generation. See S1 Fig for a pictorial description of the selection protocol.

### Bioluminescence recording and analysis

Cells were plated in white 96-well plate at a density of $20 \times 10^3$ cells/well, and after 72 hours, cells were synchronized with dexamethasone (1 μM) for 30 minutes, washed twice with PBS,

and cultured in Phenol-Red-free DMEM containing 10% fetal bovine serum, antibiotics (100 U/ml penicillin and 100 μg/ml streptomycin), and 250 μM D-luciferin (BioThema, Darmstadt, Germany). Bioluminescence was recorded at 37°C in a 96-well plate luminescence counter (TopCount, PerkinElmer, Rodgau, Germany) for up to 7 days. The same selection protocol was implemented on NIH 3T3 cells as well. Bioluminescence data were analyzed by using ChronoStar software [78]. In brief, the raw bioluminescence traces were detrended using a 24-h sliding window and the detrended data are used to fit a damped oscillator equation of the form y = A * (exp^ (−Ƴ*t)) *Cos (2πf*t + φ) + error; where A = amplitude, Ƴ = damping coefficient, t = time, f = frequency, φ = phase. The best-fit values are used to extract period, decay constant (damping), relative amplitude, and mean bioluminescence intensity (MESOR) of the oscillation.

## Estimation of broad-sense heritability

Both heritable variation ($V_H$) and nonheritable stochastic/environmental variation ($V_{NH}$) are known to contribute to phenotypic variation ($V_P = V_H + V_{NH}$) observed in populations. We estimated broad-sense heritability ($H^2$) using one-way ANOVA (outlined in S2C Fig; [34]). We chose a group of parental clones that vary in circadian period and assayed the circadian periods on $n$ subclones from each of the parental clone and used one-way ANOVA to partition the between-clone and within-clone variance. Variation between clones can stem from both heritable and nonheritable components and is obtained by mean squares between term ($MS_b$), where $MS_b = V_{NH} + nV_H$ with $n$ being the number of assayed subclones for each parental clone (S2C Fig). However, variation within clones is expected to be largely due to nonheritable factors, since the subclones of a parental clone are genetically identical and are given by mean squares within term ($MS_w$), where $MS_w = V_{NH}$. Provided that the same number of subclones ($n$) is assayed for all parental clones, heritable genetic variance $V_G$ can be calculated as $(MS_b − MS_w)/n$ and nonheritable variance $V_{NH} = MS_w$ (S2C Fig; [34]). Thus, broad-sense heritability ($H^2 = V_H/V_P$) was estimated to be $H^2 = V_H/(V_H + V_{NH})$.

## Intercycle period stability analysis

Detrended bioluminescence traces were analyzed in R using peakPick package [79] to identify peaks in the data. The duration between two consecutive peaks represents the period of that cycle and the SD of periods across multiple consecutive cycles was used as a measure of intercycle period stability. Since short-period rhythms have more cycles than long-period rhythms in a given time duration, we restricted the number of cycles considered for analysis to three to four cycles for all the time series data analyzed.

## Whole-exome sequencing and analysis

Genomic DNA was isolated using Qiagen Blood & Cell Culture DNA Mini Kit (Cat. # 13323). DNA quality was monitored on 1% agarose gels and concentration was measured using Qubit DNA Assay Kit in Qubit 2.0 Flurometer (Life Technologies, Carlsbad, CA, USA). Exomes were captured using and Agilent SureSelect Human All Exon kit V6 (Agilent Technologies, Santa Clara, CA, USA) following manufacturer's recommendations, and enriched libraries were sequenced at 50X on Illumina Novaseq (2 × 150 bp) platform by Novogene. After quality-filtering raw reads, clean paired-end reads were mapped to human reference genome (hg38) using Burrows-Wheeler Aligner [80]; the resulting BAM files were sorted using SAM tools [81] and Picard was used to mark duplicate reads [82]. GATK [83] was used for calling SNPs and small insertions and deletions (InDels). CONTRA [84] was used for CNV detection.

Variant annotation and effect prediction were performed using ANNOVAR [85] and VEP [86].

## RNA isolation and NanoString-based gene expression analysis

Cells were plated at a density of approximately $20 \times 10^3$ cells/well in 24-well plate with DMEM containing 10% fetal bovine serum and antibiotics (100 U/ml penicillin and 100 μg/ml streptomycin) and were left undisturbed without medium replacement for 144.5 h. When handled this way, the cell population was essentially arrhythmic by day 6, with relative amplitudes reducing by approximately 95%, comparing day 1 amplitude (0.44 ± 0.09; mean ± SD) with day 6 (0.02 ± 0.01; mean ± SD)—a residual amplitude too low to be biologically significant (S7A and S7B Fig). In addition, we measured *DBP* (a circadian clock gene with high-amplitude expression) expression on two different instances in representative SCL and LCL clones on day 6 by sampling cells over 28 h between 144.5 ± 14 h. We did not find *DBP* expression to be rhythmic in either of the clones (MetaCycle $p > 0.5$; S7C and S7D Fig).

At 144.5 h post cell plating, the culture medium was aspirated, and 100 μl/well iScript RT-qPCR Sample Preparation Reagent (Bio-Rad) was added on top of the cell layer and incubated at 37°C for 5 min. A sample of 3 μl was withdrawn without disturbing the cell layer and used for further downstream analysis.

A previous study of ours combined whole-genome transcriptomics with machine learning and identified genes that can serve as reliable circadian markers [87]. Based on this, we designed a 24-plex NanoString probe panel comprising 20 circadian clock and clock-associated genes along with four housekeeping genes (S1 Table). The custom-designed probes included a 3′-end biotinylated capture probe and a 5′-fluorescence-barcoded reporter probe for each gene target. Hybridization of probes and gene expression–count reading was according to the manufacturer's instructions. Raw expression data were acquired by a NanoString nCounter Digital Analyzer (NanoString Technologies), QC processed, and analyzed by nSolver. Of the 24 genes in the panel, we discarded data from one gene (*CIART*) because it failed to pass QC analysis. Data normalization involved three steps: (1) normalization by the arithmetic mean of the positive spike-in controls, (2) subtraction of the mean of all negative controls, and (3) normalization by the geometric mean of the four housekeeping genes.

For all other experiments not involving NanoString platform, total RNA was isolated with AMBION PureLink RNA Mini kit (Thermo Fisher) including an on-column DNase digest according to manufacturer's instructions. The isolated RNA was then reverse transcribed using M-MLV Reverse Transcriptase (Life Technologies), and gene expression was measured by qPCR in a CFX96 thermal cycler (Bio-Rad, Munich, Germany) using gene-specific QuantiTect primers (Qiagen).

## PCA and clustering

Log$_2$-transformed gene expression data were first subjected to Bartlett's Test of Sphericity to validate its adequacy for PCA, after which correlation-based PCA was implemented in R [88] using factoextra and FactoMineR packages [89]. Broken Stick model [47] was used to determine the number of retainable PCs. Determining the optimal cluster number is often a complication in unsupervised exploratory data analysis. Unlike many studies in biology that employ PCA to identify genes based on expression differences between known cell types (which can be used to estimate the optimal number of clusters), our study employs a panel of clones with a continuous distribution of phenotypes (period) and thus cannot be categorized trivially. Hence, we adopted two schemes for optimal cluster number determination. (1) For agglomerative hierarchical clustering, we assessed the agglomeration schedule to identify the

possible number of clusters [90]. (2) In addition, we performed k-means clustering for different values of cluster (k = 1–10) and used five different indexes—"silhouette method" [91], "elbow method" [92], "gap statistic" [93], "Calinski-Harabasz criterion value (variance-ratio method)" [94], and Bayesian information criterion (BIC; [95])—to assess the optimal cluster number. We selected the optimal cluster number based on agreement between (1) and (2). Heatmapper [96] and "dendextend" [97] were used for hierarchical clustering analysis based on "euclidean-distance" and "complete-linkage" measures [98]. "Nbclust" [99] and "mclust" [100] were used for k-means-based clustering analysis, and for all others, statistical analysis and graphing was performed using R and Prism version 8.00 for Windows (GraphPad Software, La Jolla, CA, USA, www.graphpad.com).

### RNAi-mediated gene knockdown

The GIPZ microRNA-adapted shRNA constructs used for the study were purchased from Open Biosystems and packaged into lentiviral vectors in HEK293T cells in a 96-well plate format [32]. Virus-containing supernatants were then filtered and U-2 OS cells were transduced with 150 μL of the filtrate containing 8 ng/μL protamine sulfate. The filtrate was replaced after at least 24 h with fresh medium containing puromycin (10 μg/mL). After 3 days, the transduced cells were synchronized and bioluminescence was recorded as described above.

### Cell proliferation assay and SAHA treatment

To measure cell proliferation, cells were plated in 96-well plates at two different densities (1,000 and 5,000 cells/well) and four replicates each. Starting 24 h after plating, cell proliferation was measured for 7 consecutive days by colorimetric assay with Vybrant MTT Cell Proliferation Assay Kit (Thermo Fischer Scientific, catalog #V13154). To account for errors in cell counting and plating, absorbance on day 1 was set to 1 and absorbance on consecutive days was calculated relative to that on day 1.

For experiments involving SAHA treatment, $10^3$ cells/well were plated in 96-well plates on day 0. After 24 h, culture medium was replaced with media containing 1.6 μM SAHA or DMSO vehicle control. Drug-containing medium was replaced every day for 3 consecutive days, after which cells were rinsed thrice with PBS, and fresh culture medium without drug was added on day 4. Since wells are 50% confluent on day 4 (see $IC_{50}$ estimation below), to avoid artefacts density on circadian period, cells were left untreated for 48 h and bioluminescence rhythms were recorded from day 6.

The above-described protocol was followed for estimating $IC_{50}$ value as well. Cells were treated with varying concentrations (0–100 μM) of SAHA from day 1, and cell proliferation was assayed on day 4 using the Vybrant MTT Cell Proliferation Assay Kit (Thermo Fischer Scientific, catalog #V13154) as per manufacturer's protocol. $IC_{50}$ was calculated from the resulting dose-response curve using Prism version 8.00 for Windows (GraphPad Software, La Jolla, CA, USA, www.graphpad.com; S12 Fig).

### Methylation analysis

Methylation was assessed by methylation-sensitive restriction enzyme (MSRE)-based qPCR using OneStep qMethyl-PCR Kit (Zymo Research). CpG islands spanning transcription start sites of respective genes were identified and sequences were extracted using Genome Browser (http://genome.ucsc.edu/). Primers were designed using NCBI Primer-BLAST to contain at least two MSRE sites as per manufacturer's guidelines. Two sets of primers targeting different regions of the CpG islands were tested for every gene, and one set per gene (S5 Table) was considered based on amplification efficiency and measured percent methylation for the given

region. Genomic DNA was isolated from confluent cells in a 6-well plate using Qiagen Blood & Cell Culture DNA Mini Kit (Cat. # 13323) and used for qPCR with cycling conditions as suggested in manufacturer's protocol.

## Supporting information

**S1 Fig. Graphical depiction of the selection protocol.** The selection protocol adopted for deriving the panel of short- and long-period clones used in this study is graphically represented.
(TIF)

**S2 Fig. Both heritable and nonheritable components underlie divergence in circadian period between the short- and long-period clones.** **(A)** Raw bioluminescence traces of representative clones from founding culture (dashed line), short-period (red), and long-period (blue) clonal lines. **(B)** Regression of progeny cultures' periods on mean periods of their parental cultures' periods. Each data point is an average of three to five experiments. Blue solid line is the linear regression fit with its 95% CI (green dotted line). **(C)** Pictorial depiction of variance partitioning between and within clones to estimate heritability. If a group of parental clones (P1–P6) exhibiting different circadian periods are considered and their $n$ progeny periods are assayed, the between-clone variance ($MS_b$) provides an estimate of phenotypic variation due to both heritable ($V_H$) and nonheritable ($V_{NH}$) mechanisms, whereas the within-clone variance ($MS_w$) is likely due to nonheritable/environmental variation ($V_{NH}$). Thus, these two variance components can be used to estimate heritability ($H^2$) as depicted above. Underlying data for this figure can be found in S1 Data.
(TIF)

**S3 Fig. In accordance with entrainment theory, short- and long-period clones entrain with different phases.** Entrainment profiles of a representative short- (red) and long-period (blue) clone to T24 (12 h of 37˚C and 33˚C each) (top panel) and T26 (13 h of 37˚C and 33˚C each) (bottom panel). Red and blue dots indicate peak phases on respective days, and blue-shaded boxes indicate low temperatures. Underlying data for this figure can be found in S1 Data.
(TIF)

**S4 Fig. Clonal heterogeneity in circadian period is likely a ubiquitous phenomenon.** Divergence of circadian period distributions of short-period (red) and long-period (blue) clones from a common founding culture (gray) across one assay generation for NIH 3T3 cells. Dashed black lines depict the mean of respective period distributions. The gray dashed lines extended from assay generation 1 depict mean period of the founding culture (assay generation 0) for visual assessment of the period divergence. Red arrows (short-period clone) and blue arrows (long-period clone) indicate the periods of representative clones selected for the successive assay generation. Underlying data for this figure can be found in S1 Data.
(TIF)

**S5 Fig. Clonal circadian period heterogeneity in U-2 OS cells is likely not due to polymorphisms.** mRNA expression (normalized to *GAPDH*) of **(A)** *SART3*, **(B)** *ZMAT3*, and **(C)** *GFAP* comparing three short (SCL) and long (LCL) representative clones each. SCLs and LCLs were not significantly different in their expression of the above-mentioned genes (randomized block design ANOVA, $p > 0.05$). Underlying data for this figure can be found in S1 Data. LCL, long-period clonal line; SCL, short-period clonal line.
(TIF)

**S6 Fig. Relative amplitude and damping rate are weakly heritable.** Linear regression of mean progeny values on parental values for **(A)** relative amplitude ($R^2 = 0.04$) and **(B)** damping rate ($R^2 = 0.40$). Each data point is an average of three to five experiments. Green solid line is the linear regression fit with its 95% CI (green dotted line). ****$p < 0.0001$. Underlying data for this figure can be found in S1 Data.
(TIF)

**S7 Fig. Observed differences in gene expression across 25 clones is not a phase effect. (A)** Bioluminescence traces of representative clones from our clonal panel depicting the damping of rhythm over 6 days. Blue-shaded regions indicate the time windows on day 1 and day 6 when the amplitude was measured (shown in B). Red dashed line indicates the time at which RNA was isolated for gene expression quantification (Fig 3D and 3E). **(B)** Relative rhythm amplitudes on day 1 and day 6 of the bioluminescence traces presented in (A). Error bars are SD across 30 clones. **(C)** mRNA expression of *DBP* (normalized to *GAPDH*) from an SCL and LCL sampled at 4-h intervals for 28 h on day 6, blue-shaded region in (A). To ensure reproducibility, SCL and LCL were sampled independently on two different experiments. MetaCycle analysis did not report a significant rhythmicity in either of the clones. Error bars are SD across three qPCR runs. **(D)** Average (across 28 h) *DBP* expression in SCLs and LCLs as measured on day 6. In agreement with Fig 3D, SCLs have higher *DBP* expression compared to LCLs, thus further confirming that the gene expression measured on day 6 indicates gene expression from asynchronous clones. Error bars SD across time points for the three qPCR runs. ****$p < 0.0001$. Underlying data for this figure can be found in S1 Data. LCL, long-period clonal line; qPCR, quantitative PCR; SCL, short-period clonal line.
(TIF)

**S8 Fig. Heritable variation in circadian clock genes likely underlies clonal heterogeneity in circadian period. (A)** Pearson correlation of gene expression among 19 clock and clock-associated genes assayed in our clonal panel indicates a high degree of cross-correlation between the clock genes, likely due to the interconnected molecular clock network where change in expression of one gene drives expression changes in many other genes. Because of such high correlations between the genes measured, we adopted PC analysis to identify genes contributing the most to period heterogeneity. **(B)** Variable (gene) correlation map used to visually assess how strongly a variable (gene) is correlated with a PC. In general, the lower the angle between the vector depicting the gene and the PC, the stronger the correlation of the gene with the PC. **(C-D)** Contributions of the 19 analyzed genes to PC2 and PC1 respectively. Underlying data for this figure can be found in S1 Data. PC, principal component.
(TIF)

**S9 Fig. Top five genes from principal component 2 cluster clones into three groups, whereas those from principal component 1 clusters clones into two groups.** To estimate the optimal number of clusters in our dataset, we measured five different k-mean clustering indexes (average silhouette width, WSS, gap statistic, Calinski-Harabasz value, and Bayesian information criterion) for k = 1–10 clusters. **(A-E)** Values of the above-mentioned indexes across 10 clusters generated by the five selected genes from principal component 2. **(F-J)** The same for clusters generated by the five selected genes selected from principal component 1. The red dots indicate the optimal cluster number chosen based on the respective index measure. Details of the indexes used and their interpretation can be found in the respective references (see Methods). In brief, for all indexes except WSS, the cluster number resulting in the highest value of the index was considered likely to be the optimal cluster number. For WSS, the cluster number at which the WSS plot forms an elbow joint (the magnitude of drop in

WSS values reduces thereafter) was considered as the likely optimal cluster number. When more than one optimal cluster number is observed, as in (C), (D), and (H), the decision was based on guidelines suggested by the original authors. Raw data used for analyses in this figure can be found in S1 Data. WSS, within sum of squares.
(TIF)

**S10 Fig. Clonal heterogeneity in circadian period is associated with altered expression of clock and clock-associated genes.** Trends of gene expression across clones exhibiting different circadian periods for the five selected genes from **(A)** PC2 and **(B)** PC1. $r$ = Pearson correlation coefficient and $R^2$ = goodness of linear regression fit (blue solid lines) to estimate the proportion of variance in clone period explained by variance in gene expression. $^*p < 0.05$; $^{**}p < 0.001$; $^{***}p < 0.0001$; $^{****}p < 0.00001$. Underlying data for this figure can be found in S1 Data. PC, principal component.
(TIF)

**S11 Fig. Both cell density and circadian period likely influence cell proliferation.** Assessment of cell proliferation rate across 7 consecutive days in SCLs and LCLs post plating at different starting densities—1,000 cells/well **(A)** and 5,000 cells/well **(B)** in a 96-well plate. To account for variations in cell counting and plating, absorbance values are expressed relative to day 1. Error bars represent SD across three representative SCLs and LCLs used. Underlying data for this figure can be found in S1 Data. LCL, long-period clonal line; SCL, short-period clonal line.
(TIF)

**S12 Fig. Estimation of $IC_{50}$ value for SAHA treatment.** To estimate $IC_{50}$ value for SAHA, cells were treated with varying concentrations of the drug (0–104 μM) for 3 days (see Methods), after which cell proliferation was measured as absorbance at 570 nM using Vybrant MTT Cell Proliferation Assay Kit (Thermo Fischer Scientific, catalog #V13154). From the resulting dose-response curve, $IC_{50}$ was calculated using Prism version 8.00 for Windows (GraphPad Software, La Jolla, CA, USA, www.graphpad.com). Error bars on data points represent SD ($n$ = 3). The solid dashed line is the nonlinear regression fit with its 95% CI (dotted line). Underlying data for this figure can be found in S1 Data. SAHA, suberoylanilide hydroxamic acid.
(TIF)

**S1 Table.**
(XLSX)

**S2 Table.**
(XLSX)

**S3 Table.**
(XLSX)

**S4 Table.**
(XLSX)

**S5 Table.**
(XLSX)

**S1 Data.**
(XLSX)

**S1 Text.**
(DOCX)

## Acknowledgments

The authors thank Sebastian Jäschke, Katja Schellenberg, Astrid Grudziecki, Almut Eisele, and Rebekka Trenkle for their help with single-cell cloning; Hedwig Lammert, Michael Hummel, and Barbara Koller for help with NanoString experiments; and Lucas Hille for his help with gene-knockdown and epigenetics-related experiments.

## Author Contributions

**Conceptualization:** K. L. Nikhil, Sandra Korge, Achim Kramer.

**Data curation:** K. L. Nikhil, Sandra Korge.

**Formal analysis:** K. L. Nikhil, Sandra Korge, Achim Kramer.

**Funding acquisition:** Achim Kramer.

**Investigation:** K. L. Nikhil, Sandra Korge.

**Project administration:** Achim Kramer.

**Supervision:** Achim Kramer.

**Writing – original draft:** K. L. Nikhil.

**Writing – review & editing:** Achim Kramer.

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
