## [Editor Report · Decision Letter 0]

18 Sep 2019

Dear Achim, 

Thank you again for submitting your manuscript entitled "Heritable gene expression variability governs clonal heterogeneity in circadian period" for consideration as a Short Report by PLOS Biology.

Your manuscript has now been evaluated by the PLOS Biology editorial staff as well as by an Academic Editor and I am writing to let you know that we would like to send your submission out for external peer review. Please note, however, that the outcome of our discussion of your manuscript is that we have some reservations as to the strength of analysis and evidence, especially given the studies were done in a single cell line. This will likely have to be addressed if we pursue it further for PLOS Biology after review. 

Before we can send your manuscript to reviewers, we need you to complete your submission by providing the metadata that is required for full assessment. To this end, please login to Editorial Manager where you will find the paper in the 'Submissions Needing Revisions' folder on your homepage. Please click 'Revise Submission' from the Action Links and complete all additional questions in the submission questionnaire.

Please re-submit your manuscript within two working days, i.e. by Sep 20 2019 11:59PM.

Kind regards,

Hashi Wijayatilake, PhD,

Acting Chief Editor

PLOS Biology

---

## [Decision Letter · Decision Letter 1]

16 Oct 2019

Dear Achim,

Thank you very much for submitting your manuscript "Heritable gene expression variability governs clonal heterogeneity in circadian period" for consideration as a Short Reports at PLOS Biology. Your manuscript has been evaluated by the PLOS Biology editors, an Academic Editor with relevant expertise, and by independent reviewers. I'm so sorry again for the delay while we discussed the reviews with the Academic Editor who needed some extra time.

The reviews of your manuscript are appended below. You will see that the reviewers find the work potentially interesting. However, based on their specific comments and following discussion with the Academic Editor, I regret that we cannot accept the current version of the manuscript for publication. We certainly remain interested in your study however and would be willing to consider resubmission of a revised version that addresses the reviewers' comments. The Academic Editor feels the key issues are as follows - 1) the cell type - as the cell type used is cancer derived this could be a problem for interpreting their data. The Academic Editor agrees with Reviewer 2 that sequencing is essential. We acknowledge that this is a large ask and are willing to allow extra time (see below). 2) Phase effect - we agree with Reviewers 1 (and also Reviewer 3) that phase needs to be considered in the study. 3) Perhaps most importantly, the Academic Editor agrees with Reviewer 2 that the stochastic idea is a red herring and needs to be addressed as requested.

Overall, we appreciate that these requests, which we feel are essential to support the current primary conclusions and appropriately interpret the data, represent a great deal of extra work, and we are willing to relax our standard revision time to allow you six months to revise your manuscript. We also feel it is likely the eventual revision will be more suitable as a Research Article, rather than a Short Report. Please modify the article type accordingly when resubmitting.

Please email us (plosbiology@plos.org) to discuss this if you have any questions or concerns, or think that you would need longer than this. At this stage, your manuscript remains formally under active consideration at our journal; please notify us by email if you do not wish to submit a revision and instead wish to pursue publication elsewhere, so that we may end consideration of the manuscript at PLOS Biology.

Please note that we cannot make any decision about publication until we have seen the revised manuscript and your response to the reviewers' comments. Your revised manuscript would be sent for further evaluation by the reviewers.

Your revisions should address the specific points made by each reviewer. Please submit a file detailing your responses to the editorial requests and a point-by-point response to all of the reviewers' comments that indicates the changes you have made to the manuscript. In addition to a clean copy of the manuscript, please upload a 'track-changes' version of your manuscript that specifies the edits made. This should be uploaded as a "Related" file type. You should also cite any additional relevant literature that has been published since the original submission and mention any additional citations in your response. 

Before you revise your manuscript, please review the following PLOS policy and formatting requirements checklist PDF: http://journals.plos.org/plosbiology/s/file?id=9411/plos-biology-formatting-checklist.pdf. It is helpful if you format your revision according to our requirements - should your paper subsequently be accepted, this will save time at the acceptance stage.

Please note that as a condition of publication PLOS' data policy (http://journals.plos.org/plosbiology/s/data-availability) requires that you make available all data used to draw the conclusions arrived at in your manuscript. If you have not already done so, you must include any data used in your manuscript either in appropriate repositories, within the body of the manuscript, or as supporting information (N.B. this includes any numerical values that were used to generate graphs, histograms etc.). For an example see here: http://www.plosbiology.org/article/info%3Adoi%2F10.1371%2Fjournal.pbio.1001908#s5.

For manuscripts submitted on or after 1st July 2019, we require the original, uncropped and minimally adjusted images supporting all blot and gel results reported in an article's figures or Supporting Information files. We will require these files before a manuscript can be accepted so please prepare them now, if you have not already uploaded them. Please carefully read our guidelines for how to prepare and upload this data: https://journals.plos.org/plosbiology/s/figures#loc-blot-and-gel-reporting-requirements.

If you still intend to submit a revised version of your manuscript, please go to https://www.editorialmanager.com/pbiology/ and log in as an Author. Click the link labelled 'Submissions Needing Revision' where you will find your submission record. 

Sincerely,

Hashi Wijayatilake, PhD, 

Managing Editor

PLOS Biology

REVIEWS:

Reviewer #1 (John O'Neill): 

This manuscript addresses a fascinating biological question, is nicely written, with the data describing the phenomenon being most convincing, and resulting from a well-conceived screen. Essentially, the authors ask what proportion of the variation in the fidelity of circadian transcriptional rhythms in cultured human cells is hereditable. They design an assay to identify and isolate several short and long period clones and then investigate the extent to which variation in clock gene expression accounts for period phenotype.

The authors note that their findings in later figures are somewhat preliminary but there is an issue with the interpretation of the data presented in figures 2 & 3 that I feel the authors really should address, and will require just a couple of simple experimental controls for circadian phase and cell division as described below: 

Cell division control – U2OS cells are transformed and do not fully contact inhibit i.e. they reach a steady state where the rate of cell death = rate of cell division. Since circadian period is affected by cell division, with the consensus being with faster dividing cell populations having a shorter periods (Nagoshi et al, Cell, 2004; O’Neill & Hastings, JBR, 2008; Feillet et al, PNAS, 2014; Bieler et al, Mol Syst Biol, 2014). Thus, a plausible interpretation of the data in figure 1 is that the screen has selected for clones with different doubling times such that SCL clones proliferate more rapidly than the parental line, LCL clones proliferate more slowly. A simple measurement of doubling time of the different clones would reveal whether this alternative hypothesis can be rejected or not, but either way this control is essential for understanding the inferences that can reasonably be made about circadian timing in the different clones. Indeed, if there is a reasonable correlation between cell proliferation and circadian period the authors can estimate what proportion of period heritability can be attributed to differences in cell division compared with the other variables they consider.

Circadian phase control – Perhaps the authors are not aware that a media change to fresh media containing 10% serum synchronises cells just as effectively as dexamethasone, and so the clonal lines used for RNA expression analysis in Figures 2 & 3 simply cannot be assumed to be asynchronous, as implied in the methods section. From current understanding of the molecular clock mechanism, the mRNA levels of most of the genes on the nanostring chip will change over the course of each circadian cycle. It is therefore quite plausible that differences in clock gene mRNA levels detected between the different clones result from their being at different circadian phases at the time of RNA extraction, due to their inherent differences in period described in figure 1 i.e. once 5 days have passed since the media change, the inherent differences in period will ensure that these cells are at different phases to each other and therefore have different mRNA levels. To address this potential confound, the authors simply need to take each of the clonal lines used for RNA extraction, culture them using the same protocol (including media change with 10% serum), and record bioluminescence from them for 5 days so that they know with high accuracy the circadian phase of each of the clonal lines at the time of sampling. They can then repeat some of the analyses in figure 2 to address to what extent phase differences can account for the differences in gene expression observed between clones with different circadian periods.

Both of the controls above are quick to perform (<2 weeks duration)) and will enormously strengthen the amount of confidence that can be placed in the authors conclusions i.e. they assume causality to be:

Change in clock gene expression => difference in period

But equally plausible are two competing hypotheses:

Shorter/longer cell cycle => shorter/longer circadian period => altered clock gene expression

&/or

Altered circadian period => different phases at time of sampling => altered gene expression detected

Whereas these alternate possibilities really need to be excluded, or at least addressed, for the rest of the results section and discussion to hold water.

Minor comments:

To aid the comprehension of the non-expert reader, it would be very helpful if the authors could present representative bioluminescence traces in a main figure, rather than requiring the reader to look into the supplementary data. This is because the differences in period of oscillation and bioluminescence level between different clonal populations are the phenomenon under investigation, and so it will be much easier to understand for a naïve reader to understand what was observed if they have an example in front of them. 

For the method of analysis of bioluminescence data, the authors cite a paper that is not yet published and could not be downloaded – when I clicked on the related manuscript link it simply downloaded the present manuscript again. Therefore, whilst I am quite sure there is no problem with their analyses, it is not the method of analysis was not something I was able to assess. I presume, for example, that the first 24h of bioluminescence data were excluded from analysis since this is affected by the transient response to DXM and cannot be assumed to be purely circadian regulated. Please can the authors report their analysis method in the methods section.

Figure 3a/b needs some quantification for knockdown efficacy so that the effect on period can be compared with the data presented in Supp Fig6.

Fig2d shows a very small effect size and is only just significant i.e. it could be a fluke. A couple of additional HDAC inhibitors showing the same differential effect on period would be more convincing. 

Page 4, first line. The assumed mutation rate for human cells is not appropriate for these highly transformed cancer cells, which (almost by definition) will be defective in one or more DNA repair mechanisms. Indeed since U2-OS cells are aneuploid (Mukherji M, et al. 2006, PNAS), the authors should check whether any of the genes in PC2 or PC1 are on the same chromosome, as a change in chromosome copy number could also account for their observations - in principle at least.

--

Reviewer #2: 

The manuscript entitled "Heritable gene expression variability governs clonal heterogeneity in circadian period" by Nikhil et al. is an imaginative approach to defining major determinants underlying the period of cellular circadian clocks. The authors reach conclusions that may surprise many chronobiologists, and is a worthy addition to the literature, both in terms of approach and results obtained.

Major points:

1. The results with SAHA in Fig. 3d show such a small period difference that (even if statistically significant) are unlikely to be biologically significant/informative as to the period differences between the "short" and "long" clones. Please see Point # 4 below.

2. DISCUSSION: in Baggs et al. 2009, siRNA knockdown of Bmal2 had no effect on period. Do the authors have an explanation for the difference between their results (e.g., with the "intermediate clone") and those of Baggs et al. ?

3. General point: in the opinion of this reviewer, the authors' strategy to emphasize "heritable mechanisms to drive period-heterogeneity over entirely stochastically driven heterogeneity" is a so-called "red herring" (i.e., misleading or distracting). Based on what we know about circadian clocks, OF COURSE it will be heritable rather than stochastic. But for the authors to prove this point, they need to do exosome sequencing and methylome sequencing. In other words, the authors need to prove that this is a heritable effect by demonstrating either epigenetic marks and/or genetic polymorphisms (again, see Point # 4 below).

4. The authors' arguments against genetic polymorphisms being responsible for the effects they observe (starting line 263) are unconvincing, especially given that the changes they observe with SAHA are small and unlikely to be biologically significant (especially for period shortenings). U2OS cells are derived from osteosarcoma and therefore like most cancer cells are probably hypermutable. The authors appear to be arguing rather than doing the obvious experiment, which is to sequence the four major contributing genes that the PC analysis identifies: NR1D2, DBP, ARNTL2, and BHLHE40 (they can do at least exome sequencing). If they find no genetic polymorphisms, then they should undertake bisulphite sequencing (methyl-Seq) to determine if their preferred "epimutation" interpretation is supported by experimental measurements rather than by speculation.

Minor points:

(a) The description of the Principal Component Analysis is not understandable to me. I'm hoping that one of the other reviewers understands this analysis and can comment authoritatively upon it. If the other reviewers have a similar problem to mine, then the authors need to explain better what is really being compared here.

(b) Line 49: there is no Leise et al. 2016 in the reference list. Do the authors mean Leise et al. 2012 ?

(c) In some of the figures (e.g., lines in Fig. S2, dots in Fig. 2d, and even the histograms in Fig. 3a/b), the difference between blue and green is hard to see. Please change some of the line forms (i.e., dashed vs. solid) or use black instead of blue or green.

(d) Fig. S3c: correct typo in abscissal label "damping." 

(e) Fig. 2b: explain the Pearson correlation coefficient here. Explanation of this panel is inadequate in the legend and non-existent in the Methods. What is being correlated here?

(f) DISCUSSION: please discuss whether the authors believe this period heterogeneity to be generalizable in vivo, or is only observable with cancer cells. If the authors believe this to be generalizable to normal cells in vivo, please discuss whether the authors expect there to be a mosaic of circadian periods in the body? If that is true, then perhaps the approach used by other labs of isolating fibroblasts from the body and comparing the fibroblast periods in vitro with chronotype (as has been reported by other groups) is not a valid approach ?

--

Reviewer #3: 

This is an outstanding paper describing a clever, hypothesis-directed experiment The results are satisfying to see and they will launch a multitude of experiments that will on one hand elaborate cellular principles of the circadian clock and on the other hand molecular mechanisms. There are many gems in this paper that remain undeveloped. Given that it is a short report, perhaps this is intentional. One important but obscure principle that is elaborated is that short periods are not the mechanistic opposite of long ones. By showing the differential effect of SAHA, this concept is clear and should help us to understand how this property is regulated and what it is for.

Here are comments which are all minor:

1. As it reads:

11. A ubiquitous feature of circadian clocks across life forms is its organization as a network of 

12. 13 coupled cellular oscillators. 

The first sentence of the abstract is not correct. How do we know that liver cells are coupled versus just oscillating with an accurately similar period? What about cyanobacteria? Is there evidence that they couple?

2. It is important to be completely thoroughly clear on what is being described. The authors confuse the reader when they start the results section with:

76. Is the variation in period among individual circadian oscillator cells just due to intrinsic and/or 

77. 79 extrinsic stochastic noise? 

Stochastic was defined but not intrinsic and extrincis and these distinctions are not used much after this point. Further, stochastic is not used much in the discussion. Maybe is it not necessary?

3. This:

76. As expected, we observed a distribution of circadian periods 

is a novel finding for U2OS. There is a lot of information in here. Is the gentle curve of expression of LUC in these cells attributable to the period distribution or to gentle kinetics of expression? We have seen this distribution of periods in SCN neurons and which other cells?

4. The use of significance as a word:

76. the periods of short and long period clonal lines (SCL and LCL) 

77. 92 significantly diverged from each other 

Indicate the statistical significance here. The following sentence indicates that the divergence is not significant after generation 2. The use of significance must be cleaned up. Why give the values in one place and not the other. Also I am confused by the statistics used here. Anova picks up differences and here the p is very low. Why is this used to justify lack of divergence?

5. How was the number of cells controlled for?

97. We observed a positive correlation of bioluminescence intensity 

6. It might be good to check in with Baggs et al who did extensive siRNA studies:

97. correlation of relative amplitude with period was negative 

7. The wording here is vague:

97. We reasoned that mean 

98. 114 bioluminescence intensity can, in-principle serve as a proxy for the average expression level 

Some people read expression “how much is there at a given time” others read it “how much is produced at a given moment”. This should be clarified because here it is important to refer to the first definition.

8. Here, I believe that the cells probably do not respond to the selection protocol.

202. thus resulting in a directional response (divergence of short and long 

203. 204 period clones from the founding culture) to our selection protocol 

9. It would be elegant if the authors could refer to old observations on Neurospora mutants (the long period frq7 is more robust, higher amplitude than wt; the short frqs is lower. In flies, the short period pers is more robust than wr, the long mutant perl is less.

202. long-period clones often exhibited higher 

203. 218 bioluminescence intensity 

10. Along the same lines, ANRT2 expression is dependent on BMAL1/ARNT1. It would be nice to discuss this since the ARNL2 levels are so important for the conclusion here. 

11. For bioluminescence intensity: how do we know for sure that after 5 d in cluture, the cells would be non-synchronised?

12. I don’t completely agree with this proposition. When one looks at the multiple loops, mutations in e.g. the revERBs for instance look like a more devastating loss to the clock than Per2 for instance. I think it is only historical that PER CRY is core and the others are auxiliary. Per Cry was described first. 

224. This reinforces the idea that while persistence of circadian oscillation requires 

225. 239 a functional core clock loop involving negative feedback by the PER-CRY family, modulation of 

226. 240 clock period might be governed by interaction between multiple loops coupled by E-box 

227. 241 associated transcription factors 

13. Grammar: are observed

228. responses (Hormesis) is observed 

14. Below, is an interesting proposition. The authors might report the percent rhythmic clones in each of the cloning steps. One might expect more non-rhythmic clones or not if the below proposal is correct.

245. the network might 

246. 259 also harbour hormesis-based mechanisms which impose constraints on the range of period that 

247. 260 the circadian clock can exhibit 

15. Important here to specifiy if it is an order of or orders of?

267. are observed to be order of 

268. 282 magnitude higher than DNA mutation rates 

16. Please report the method of cloning. FACS? LDA?

311. were plated as 

312. 318 single-cell clones in 96-well ‘parent plates’ and grown to confluency. 

17. There is an elephant in the room in that the relationship of phase and period is never discussed in the results section. Might add a nice dimension.

---

## [Editor Report · Decision Letter 2]

1 Jun 2020

Dear Achim,

Thank you for submitting your revised Short Report entitled "Heritable gene expression variability and stochasticity govern clonal heterogeneity in circadian period" for publication in PLOS Biology. I have now obtained advice from the Academic Editor who has evaluated your revision. We're delighted to let you know that we're now editorially satisfied with your manuscript. The Academic Editor was very pleased with the exome sequencing and the experiments on entrainment in the different clones with different periods, and we appreciate the addition of another cell line, which is important, and the data showing that it is not a phase but period effect. My only remaining questions are the following:

- Regarding the figure and table in the response to reviewers file - if this is not already present, and discussed, in some way in the main manuscript, or supplement, please consider adding it as it may help readers who have similar questions to the reviewers.

- Data - your Data Availability Statement currently says "Data are from our study whose authors may be contacted at achim.kramer@charite.de" - this does not comply with the PLOS Data Policy, which can be found here: https://journals.plos.org/plosbiology/s/data-availability

Please see below for more details about the Data Policy and related requests.

Before we can formally accept your paper and consider it "in press", we also need to ensure that your article conforms to our guidelines. A member of our team will be in touch shortly with a set of requests. As we can't proceed until these requirements are met, your swift response will help prevent delays to publication. Please also make sure to address the data and other policy-related requests noted at the end of this email. IMPORTANT: Please also make sure to address the data and other policy-related requests noted at the end of this email.

*Copyediting*

*Published Peer Review History*

*Early Version*

*Submitting Your Revision*

Sincerely,

Hashi Wijayatilake, PhD, 

Managing Editor

PLOS Biology

DATA POLICY:

Figs. 1ABC, 2A-D, 3A-E, 4A-E, S2AB, S3, S4, S5, S6, S7, S8, S9, S10, S11, S12

---

## [Editor Report · Decision Letter 3]

13 Jul 2020

Dear Dr Kramer,

On behalf of my colleagues and the Academic Editor, Samer Hattar, I am pleased to inform you that we will be delighted to publish your Research Article in PLOS Biology. 

Early Version

PRESS 

Kind regards,

Alice Musson

Publishing Editor, 

PLOS Biology

on behalf of

Ines Alvarez-Garcia,

Senior Editor

PLOS Biology